# Influencing Factors of Undermet Care Needs of the Chinese Disabled Oldest Old People When Their Children Are Both Caregivers and Older People: A Cross-Sectional Study

**DOI:** 10.3390/healthcare8040365

**Published:** 2020-09-25

**Authors:** Qilin Zhang, Yanli Wu, Erpeng Liu

**Affiliations:** 1Center for Social Security Studies, Wuhan University, Wuhan 430072, China; qilinzhang@whu.edu.cn (Q.Z.); yanliwu@whu.edu.cn (Y.W.); 2Institute of Income Distribution and Public Finance, Zhongnan University of Economics and Law, Wuhan 430073, China

**Keywords:** undermet care needs, Chinese disabled oldest old people, older children, influencing factors

## Abstract

We examined the influencing factors of the undermet care needs of the Chinese disabled oldest old people when their children are both caregivers and are themselves older people. Data were obtained from a cross-sectional survey: the Chinese Longitudinal Healthy Longevity Survey (CLHLS) in 2018. The study participants included 1617 disabled oldest old people whose primary caregiver were their children or children-in-law and were aged 60 years and over. The results showed that the prevalence of undermet needs remained high, with 49.6% disabled oldest old people reporting undermet care needs. Binary logistic regression analysis revealed that living in a rural area (OR = 1.309, 95% CI = 1.133–1.513) and a higher frailty index (OR = 1.103, 95% CI = 1.075–1.131) were significantly positively associated with higher odds for undermet care needs, while a higher annual household income (OR = 0.856, 95% CI = 0.795–0.923), more financial support from children (OR = 0.969, 95% CI = 0.941–0.997), higher care expenditures (OR = 1.044, 95% CI = 1.002–1.088), better caregiver’s performance (OR = 0.282, 95% CI = 0.196–0.407) and sufficient income to pay for daily expenses (OR = 0.710, 95% CI = 0.519–0.973) were significantly inversely associated with higher odds for undermet care needs. This evidence suggests the importance of policies to establish a community-based socialized long-term care system and supporting family caregivers of the disabled oldest old people.

## 1. Introduction

Population aging is rapidly accelerating worldwide, especially in China. By 2019, the number of Chinese older people over the age of 60 years had reached 254 million, accounting for 18.1% of the total population [1]. It is estimated that by 2050, the number of older people over the age of 60 years will increase to a peak of 488 million in China, representing 35.6% of the total population [2]. At the same time, due to the increase in life expectancy, the trend of aging is also more obvious in the oldest old people. In the past 10 years, the population of Chinese older people aged 80 years and older has been growing at an average annual rate of 4.7%, which is significantly faster than the growth rate of older people aged 60 years and older [3]. In 2013, the number of Chinese older people over the age of 80 years had reached 22.6 million. It is estimated that by 2050, this number will increase 4-fold, reaching 90.4 million—becoming the world’s largest group of oldest old people [4].

China is an unhealthy aging society, and its disabled older population is developing rapidly [5,6,7]. According to the Fourth Sampling Survey on the Living Conditions of Urban and Rural Older People in China, the number of partially disabled and completely disabled older people has reached 40.63 million [8]. It is estimated that by 2030 and 2050, the number of older people with disabilities in China will further increase to 61.68 million and 97.5 million, respectively [9]. At the same time, the oldest old people are mostly unhealthy. Due to the deterioration of their physical functions and chronic disease, the probability of suffering from disability and dementia among the oldest old people is significantly higher than those of younger older people aged 60 to 79 years [10,11,12]. Moreover, the proportion of older adults who cannot take care of themselves gradually increases with age, especially in the period of 80 to 90 years old, when the proportion of older adults who cannot take care of themselves increases the fastest [13,14]. Therefore, the demand for care is the strongest among the disabled oldest old people in China.

The care of the oldest old people in China is still provided by families, which is a traditional and strong inertial old-age care model [15,16,17,18]. At the same time, the probability of widowhood is high among the oldest old population [19,20,21]; therefore, their children or children-in-law become their main body of family care [22,23,24,25]. The reasons for this phenomenon are as follows: First, the economic status of the oldest old people is poor [26], and they cannot afford the care services provided by the market. Influenced by the long-term low wage policy, the oldest old people have few savings. Additionally, due to China’s undeveloped pension system, most older people have very low pensions, with an average level of 100 to 200 RMB (Ren Min Bi, Chinese Yuan, about 14.64 to 29.28 dollars in September 2020) per month [27,28]. The proportion of older adults aged 80 years and older receiving the old age allowance is 13.64% [29], and the old age allowance in most regions is only 100 RMB (about 14.64 dollars in September 2020) per month [30]. Second, the level of public care services provided by the government is low, and the capacities of the market and social organizations to provide care services are at the initial stages [31,32]. Thus, it is difficult for older people to obtain formal care service support. Finally, the current generation of the oldest old people were born before the 1940s and were greatly influenced by the traditional culture of filial piety [33,34,35]. It is believed that caring for older parents is an obligation and has traditionally been highly valued by children [36,37]. People take "living in a nursing home" as a manifestation of a lack of filial piety, which is not conducive to family solidarity. Therefore, family care, especially by children, has become the main type of care for the oldest old people in China. At the same time, in the context of rapid aging, most children of the Chinese oldest old have entered the stage of aging. Providing care for the oldest old parents by their older children is becoming one of the care models and will be increasingly common in China [24,38].

The outbreak of the COVID-19 had a significant impact on the care of Chinese older people, especially the family care model. The Ministry of Civil Affairs, in charge of older people’s care services, had issued some policy support and guidance for institutional older people care, such as “Guidelines for Prevention and Control of COVID-19 Epidemic High-risk Areas and Infected Older People Care Institutions”. However, no corresponding policy support measures have been introduced for older people in family care. The impacts are not limited to older people cared by family members. Those older people suffering from chronic diseases could not get medicines in time. Moreover, older people were in a state of panic, restricted at home due to their inability to use information equipment proficiently.

In the context of accelerating aging in China, our study is the first to explore the care model of older children taking care of their oldest old parents. In this care model, whether the care provided meets the needs of the disabled oldest old people and its influencing factors should be highly valued. In previous studies, many disabled older people living with family members reported that the assistance they received did not fully meet their needs [24,39], and these people were viewed as having undermet care needs, i.e., they needed more care due to the insufficient amount of assistance they received [40]. Similar to the unmet needs of receiving no help, undermet care needs of receiving inadequate personal assistance in ADL (activities of daily living) are serious threats to the health status and life quality of older people with disabilities [40], including more limited ADL, emergency-room visits and hospitalizations [41,42,43], increased psychological stress [44], and higher rate of mortality [45]. Therefore, identifying the factors associated with the undermet care needs of the disabled oldest old people and taking appropriate interventions are particularly important.

While previous studies have explored the influencing factors for undermet care needs of older people, some limitations exist. First, most studies are conducted in developed countries, with only a few studies conducted in China, and the data in these studies were too old to reflect the present situation of China [39,46], especially in the context of the accelerating aging and increasing number of older people. Second, the sample of the disabled oldest old people (age 80+), who tend to need the most care, was limited and therefore under-represented in previous studies. Third, previous studies on undermet care needs of Chinese older people mostly used Andersen’s behavioral model of health services use [47,48,49], which may not fully consider the cultural characteristics and traditional customs of care in Chinese older people. Last, but most important, the care sources provided were mostly from younger family members in previous studies. Old people, especially the older children as caregivers, had not be considered. Therefore, in this study, we aimed to use the latest nationwide data to explore the influencing factors for the undermet care needs of the Chinese disabled oldest old people when their children are both caregivers and older people themselves.

## 2. Materials and Methods

### 2.1. Date Source and Study Population

This study used data from the Chinese Longitudinal Healthy Longevity Survey (CLHLS), a nationally representative survey jointly performed by the Center for Healthy Aging and Development Studies at Peking University and Duke University. The CLHLS aims to understand the health status and associated social, behavioral, and biological factors among Chinese older people. As currently the largest survey in the world in terms of health and longevity, a baseline survey was conducted in 1998, followed by seven waves of surveys in 2000, 2002, 2005, 2008, 2011, 2014 and 2018 from 22 sample areas in 31 provincial administrative units, which constituted 85% of the total population and covered East, Central West and Northeast China. The survey randomly selected approximately one-half of the counties in the 22 provinces as primary survey units. Additional details, such as the sampling design, sampling weight and assessment of data quality, could be found in previous studies [6,7,10,25,50].

We used the latest cross-sectional data of CLHLS in 2018, which consist of 15,874 Chinese older people. After filtering for missing values, outliers, etc., our analysis included 1617 disabled oldest old people aged 80 years and over whose primary caregivers were children (sons, daughters) or children-in-law (son-in-law, daughter-in-law) aged 60 years and over. The characteristics of the participants are presented in Table 1. The participants included 387 men and 1230 women. The mean age of the participants was 98.71 years (SD = 5.18 years), with 97.54 years (SD = 5.14 years) for men and 99.07 years (SD = 5.14 years) for women. Only 75 (4.67%) participants had spouses, while 1531 (95.33%) of participants did not. The average education years of the participants were approximately 1.41 years, indicating that they had low levels of education. There were 374 (23.13%) participants living in a city, 529 (32.71%) participants living in a town and 714 (44.16%) living in rural areas. The mean frailty index of the oldest old people was 0.40.

**Ethical approval:** The study was approved by the institutional review board at Centre for Social Security Studies of Wuhan University (SSRWU: EP20200701).

### 2.2. Measures

#### 2.2.1. Disabled Oldest Old People

Disabled oldest old people are defined as disabled older people aged 80 years and over in this study. Following previous studies [6,7], the ADL disabilities of the oldest old people were measured using the Katz scale [51], which gives a total score (ranging from 6 to 18) of six aspects in daily living disability, including bathing, dressing, bathroom use, indoor transferring, continence and feeding. Depending on the independence of older individuals in completing these activities, they were given a score of 1 (complete independence), 2 (partial independence) or 3 (complete dependence on others), with higher scores indicating a severe daily living disability. The oldest old people who had at least one ADL disability (an ADL score over 6) were classified as disabled oldest old people with long-term care (LTC) needs.

#### 2.2.2. Older Children

In the CLHLS, participants were asked: ‘‘who is the primary caregiver when you need assistance in the six activities of bathing, dressing, toileting, transferring, continence and feeding?’’ The answer included 12 categories: spouse, son, daughter-in-law, daughter, son-in-law, son and daughter, grandchildren, other relatives, neighbors, social service, housemaid, and nobody. The older children in our study were the primary caregivers of the disabled oldest old people, including five categories: son, daughter-in-law, daughter, son-in-law, son and daughter, and they all were aged 60 years and older.

#### 2.2.3. Undermet Care Needs

Undermet care needs were usually included in unmet care needs. Unmet care needs occur in LTC, when a person with disabilities for which help is needed is unavailable or insufficient [52]. Followed by the previous study, absent and insufficient assistance in personal activities of ADL were named unmet and undermet care needs, respectively [40,46], and we use the term “undermet care needs” to define insufficient assistance in personal activities of ADL among the disabled oldest old people in this study. The undermet care needs of the disabled oldest old people were measured based on individual’s self-reported response to the question, “Do you think the care you received could meet your needs?” Responses to this question fell into three categories: not met, partially met and fully met. In this study, both the disabled oldest old people who responded partially met and not met were classified as having undermet needs, following the practices of previous research [39,47]. This categorization allowed us to dichotomize the responses into two categories: fully met care needs (coded as 0) and undermet care needs (coded as 1).

#### 2.2.4. Influencing Factors of Undermet Care Needs

According to a review of previous research, the influencing factors of the undermet care needs of the disabled oldest old people were divided into the following categories in this study: (1) sociodemographic characteristics, including sex, age, education, marital status, place of residence and frailty index [46,53,54,55,56]. The frailty index was composed of eight parts, including the ADL, IADL (instrumental activities of daily living), objective physical performance, cognitive function, chronic disease, self-rated and interviewer-rated health, visual and hearing impairment, and number of times suffered from a serious illness in the past 2 years, which ranged from 0 to 1, with higher scores indicating weaker physical and mental functions of the oldest old people [57,58]. It should be pointed out that the Chinese version of MMSE (Mini-mental State Examination) is mainly used to measure the cognitive function of the older people, which is considered to be effective and applicable [6,12]; (2) family endowment, also called family resource, includes the human resources, economic resources, material resources, and social network resources which is the family actually or potentially owns. In our research, the measurement of family endowment mainly includes the number of cohabitating family members [47,59] and annual household income [60,61]. Previous studies have shown that the better the family endowment, the more care resources older people can obtain and the lower their undermet care needs; (3) intergenerational relationship, including children’s financial support [62,63,64], care expenditure [65,66], caregiver’s performance [46,47] and the frequency of sharing feelings with the caregiver [67,68] and (4) economic bargaining power, including whether the income was sufficient for paying for daily expenses or not [47,69], having real estate in one’s own name or not and receiving the public old age pension or not [66,70]. Descriptive statistics of the variables are shown in Table 1.

### 2.3. Statistical Analysis

The data were analyzed using Stata (Stata version 14.0 for Windows, StataCorp LP, College Station, TX, USA). We present descriptive statistics. The results are expressed as the mean (standard deviation) for continuous variables and the number (proportion) for categorial variables. The undermet care needs and sociodemographic characteristics of the disabled oldest old people were analyzed using t-test and Chi-square test. Binary logistic regression analysis was used to investigate the association between undermet care needs and the four categories of influencing factors.

## 3. Results

The sociodemographic characteristics of the disabled oldest old people according to undermet care needs in this study are shown in Table 2. Of the 1617 participants, 815 (50.4%) reported fully met care needs and 802 (49.6%) reported unmet care needs. Among the sociodemographic characteristics, there was a statistically significant difference in terms of education (χ^2^ = 7.716, *p* = 0.021) and place of residence (χ^2^ = 23.487, *p* < 0.001). Mean frailty index was higher for the undermet care needs group (t = −9.165, *p* < 0.001). There were no significant differences between groups in terms of sex, age or marital status.

We employed binary logistic regression analysis to investigate the association between undermet care needs of disabled oldest old people and the four categories of influencing factors. The results are shown in Table 3. Of the sociodemographic characteristics, living in a rural area (OR = 1.309, 95% CI = 1.133–1.513) and a higher frailty index (OR = 1.103, 95% CI = 1.075–1.131) were significantly positively associated with higher odds for undermet care needs of the disabled oldest old people. However, sex, age, education level and marital status were not significantly associated. In terms of family endowment, a higher annual household income (OR = 0.856, 95% CI = 0.795–0.923) was significantly inversely associated with higher odds for undermet care needs, while the number of cohabitating family members was not significantly associated. In terms of intergenerational relationships, more children’s financial support (OR = 0.969, 95% CI = 0.941–0.997), higher care expenditure (OR = 1.044, 95% CI = 1.002–1.088) and better caregiver’s performance (OR = 0.282, 95% CI = 0.196–0.407) were significantly inversely associated with higher odds for undermet care needs of the disabled oldest old people. However, the frequency of sharing feelings with their caregiver was not significantly associated. In terms of economic bargaining power, income sufficient for paying daily expenses (OR = 0.710, 95% CI = 0.519–0.973) was significantly inversely associated with higher odds for undermet care needs of the disabled oldest old people, while having real estate in one’s own name and having public old age insurance were not significantly associated.

## 4. Discussion

### 4.1. Main Findings and Opinions

With the rapid increase of China’s aging population, the care model for older children taking care of their oldest old parents is increasingly prevalent. To the best of our knowledge, our study is the first to explore the care model of older children taking care of their oldest old parents. In addition, as opposed to previous studies focused on factors related to undermet care needs [47,48,49], which were mainly based on the theory of Andersen’s behavioral model of health services use; our analysis of the undermet care needs was based on the background of Chinese traditional filial piety culture and the strong inertial family care model, which could fill a research gap in this field. Further, our study used nationwide data from 2018 of Chinese disabled oldest old people, which could better reflect the latest situation of undermet care needs among the oldest old population in China.

The results of our study showed that, in terms of sociodemographic characteristics, living in rural areas was associated with having more undermet care needs, which was also found in previous studies [46]. In China, due to the substantial disparities in terms of economic development level, medical, and care resources between rural and urban areas, older children as caregivers in rural families provided fewer care resources than those in urban families; therefore, rural disabled oldest old people reported more undermet care needs. Our study also found that poor health status as indicated by a higher frailty index was associated with having more undermet care needs, which was supported by previous studies [47,64]. We used the frailty index to comprehensively measure the health conditions of the disabled oldest old people, which could better explore related health factors. Further, our study found that gender, age, education level and marital status were not significantly associated with undermet care needs of the disabled oldest old people. Previous studies on sociodemographic factors, including gender, age, education level and marital status, produced mixed results [24,41,53,54].

In terms of family endowment, we found that a lower annual household income was associated with having more undermet care needs among the disabled oldest old people, which agreed with previous studies [71,72]. However, in contrast with findings of previous studies [47,59], we found that the number of cohabitating family members was not associated with undermet care needs. One explanation for this discrepancy was that caring for the disabled oldest old people is both a time-intensive and labor-intensive activity requiring substantial amounts of time and patience, and it is possible that multiple cohabitants may shirk the responsibility of providing care, thereby not necessarily contributing to lowering undermet care needs. Another explanation is that since multiple generations live together, the care needs of the older people may not be able to be satisfied when family resources are limited and the older people have less power in the distribution of family resources.

In terms of intergenerational relationships, more children’s financial support, more care expenditure and better caregiver’s performance were inversely associated with having more undermet care needs, which was supported by the findings of previous studies [46,47,59,66]. To a certain extent, the care expenditure reflects the explicit cost in the process of family care for Chinese disabled oldest old people. The more care expenditure, the more resources put into care process, and the lower the undermet care needs of the disabled oldest old people. As for the care performance, it includes both the caregiver’s objective care behavior and the caregiver’s subjective attitude. Therefore, it is a more comprehensive concept. Better care performance means that the caregiver’s behaviors are more in line with the care needs of the older people, and there are more love and positive emotions involved in the care process. However, the frequency of sharing feelings with the caregiver was not associated with undermet care needs. The explanation for this finding may be that regular emotional communication with older children does not necessarily/always bring about improvements in care among disabled oldest old people. Financial transfers matter the most. On the one hand, more children’s financial support and care expenditure means more care resources are put into the care by older children, and therefore, there are fewer undermet care needs. On the other hand, older children may be more concerned about whether they can obtain financial compensation or rewards from their oldest old parents, which will motivate their enthusiasm for providing care and improve the quality of care [21,63,66]. 

In terms of the economic bargaining power of the oldest old people, their income being insufficient to pay for their daily expenses was associated with having more undermet care needs, consistent with the findings of previous studies [46,54]. However, having real estate in one’s own name and having public old age insurance were not associated. There are possible explanations for these two factors. First, the Chinese oldest old people had less real estate. In the present study, we found that more than 80% of the oldest old disabled people did not own real estate. Second, influenced by the traditional concept of inheritance, real estate will be naturally bequeathed to their children. Thus, it is difficult for the oldest old people to mobilize their children’s enthusiasm for providing care through house property rights. That is also the reason why a house for pension program (Housing Reverse Mortgagin) could not be promoted in China [73,74]. In addition, the public old age pension of the oldest old people is low; whether there was a pension had little effect on the economic bargaining power of the oldest old people; therefore, it could not mobilize their children’s enthusiasm for care and was not associated with undermet care needs.

### 4.2. Limitations and Future Works

The present study has several limitations. First, the undermet care needs of the disabled oldest old people were evaluated with only one subjective item; it is possible that self-rated undermet care needs can be used to measure the relationship between the caregiver and care recipient, rather than true willingness. Second, due to the constraints of the data, this study lacked data on care-resource factors, such as care intensity, and primary caregiver-related information, such as education, health status, employment and income. At the same time, caring is a two-way interactive process, and the attitude and assessment of caregivers regarding care needs is also important, which was not included in our study. Future studies of the care model of older children caring for their oldest old parents should focus more attention on older children caregivers. Third, our study only used one cross-sectional dataset, which could not reflect changes in undermet care needs. Although we tried to use longitudinal data, there were too few samples after matching two or three waves of CLHLS data. Additionally, there were few nationwide census data on the oldest old people in China. Future studies should further expand the sample size or develop a nationwide census for the Chinese oldest old people to make the sample information more abundant.

## 5. Conclusions

Given that family care still has a strong institutional inertia and cultural identity, the care model of older children taking care of their oldest old parents is still reasonable but faces some challenges. Due to the aging population structure, the increasing geographical mobility of the population, the increase in the employment rate of women, and the weakening of traditional values [31,32,39], the availability of potential family caregivers is declining. First, declines in both mortality and fertility have not only accelerated population aging but have also decreased the size of families. Second, increased geographical mobility means fewer family members live near to care for older people. The impact of these changes has been shifting family structures from the traditional extended family to the nuclear family, decreasing the level of available intergenerational support. Third, as females are the main family caregivers, an increasing number of women entering the labor force means they are less available to be primary caregivers in their family. Finally, modernization and individualization have eroded the core traditional value—filial piety. All these challenges have contributed to the decline in the number of available caregivers and have made the informal older-care system even more vulnerable.

Therefore, there are some policy implications of care for disabled oldest old people. First, it is crucial to establish a comprehensive, sustainable, available and affordable long-term care system to increase formal services, such as paid home service and community-based care services. Second, given that the Chinese long-term care insurance is still in the pilot stage, it is important to promote long-term care insurance as soon as possible and to offer nursing assistance to those in need. In addition, because economic status is associated with undermet care needs, it is of great significance to provide financial assistance to the oldest old people to enhance their economic accessibility to care services, particularly those residents in rural areas where undermet needs are more prevalent and female oldest old people who live especially long lives. In particular, the payment standard of old age pension and allowance should be improved to provide more financial support. Third, the role of family caregivers and relevant support policies for them should be highly valued. The family caregiver support policy should be embedded in a series of policies in terms of population, family, fiscal and taxation. Additionally, it is necessary to focus on programs that provide care skills training, respite services, and psychological counseling for family caregivers and to try to pilot a family caregiver allowance.

## Figures and Tables

**Table 1 healthcare-08-00365-t001:** Descriptive statistics of the study sample.

Variables		Measurement	n (%)/mean (SD)
Sociodemographic characteristics	Sex	Female = 0	1230 (76.07%)
		Male = 1	387 (23.93)
	Age	Continuous measurements	98.71 (5.18)
	Education (years)	No formal education = 1	1116 (69.02%)
		1 to 6 = 2	415 (25.66%)
		7 or more = 3	86 (5.32%)
	Marital status	Without spouse = 0	1531 (95.33%)
		Have spouse = 1	75 (4.67%)
	Place of residence	City = 1	374 (23.13%)
		Town = 2	529 (32.71%)
		Rural = 3	714 (44.16%)
	Frailty index	Continuous measurements	0.40 (0.11)
Family endowment	Number of cohabitating family members	Continuous measurements	2.58 (1.67)
	annual household income	Natural logarithm after continuous measurements	9.98 (1.62)
Intergenerational relationship	Children’s financial support	Natural logarithm after continuous measurements	5.57 (3.62)
	Care expenditure	Natural logarithm after continuous measurements	3.99 (2.62)
	Caregiver’s performance	Unwilling or Without patience = 0	193 (11.94%)
		Willing =1	1424 (88.06%)
	Share feelings with the caregiver	Never = 1	135 (8.35%)
		Sometimes/ Often = 2	132 (8.16%)
		Always = 3	1350 (83.49%)
Economic bargaining power	Income sufficient for paying daily expenses	No = 0	241 (14.90%)
		Yes = 1	1376 (85.10%)
	Have real estate in own name	No = 0	1311 (81.08%)
		Yes = 1	306 (18.92%)
	Public old age pension	No = 0	1174 (72.60%)
		Yes = 1	443 (27.40%)

**Table 2 healthcare-08-00365-t002:** Sociodemographic characteristics according to undermet care needs.

Variables		Undermet Care Needs (n = 802)	Fully Met Care Needs (n = 815)	t or χ^2^	*p*-Value
		n (%)/Mean (S.D.)			
Sex	female	618 (50.24)	612 (49.76)	0.858	0.354
	male	184 (47.55)	203 (52.45)		
Age	80 to 90	73 (56.59)	56 (43.41)	4.810	0.090
	91 to 100	384 (47.29)	428 (52.71)		
	>100	345 (51.04)	331 (48.96)		
Education (years)	No formal education	574 (71.57)	542 (66.50)	7.716	0.021
	1 to 6	196 (24.44)	219 (26.87)		
	7 or more	32 (3.99)	54 (6.63)		
Marital status	have no spouse	759 (49.58)	772 (50.42)	0.071	0.790
	have spouse	36 (48.00)	39 (52.00)		
Place of residence	urban	147 (39.30)	227 (60.70)	23.487	<0.001
	town	264 (49.91)	265 (50.09)		
	rural	391 (54.76)	323 (45.24)		
frailty index		0.43	0.38	−9.165	<0.001

**Table 3 healthcare-08-00365-t003:** Binary logistic regression analysis of the influencing factors for undermet care needs in the disabled oldest old people.

Variables	Odds Ratio	SE	z-Values	*p*-Values	95% CI
Sex	1.195	0.166	1.28	0.201	0.910–1.568
Age	0.988	0.011	−1.12	0.261	0.967–1.009
Education	0.911	0.096	−0.89	0.376	0.742–1.119
Marital status	0.767	0.203	−1.00	0.316	0.457–1.288
Place of residence	1.308	0.096	3.65	0.000	1.132–1.510
Frailty index	1.102	0.014	7.43	0.000	1.074–1.130
Number of cohabitating family members	1.011	0.033	0.32	0.751	0.947–1.078
Annual household income	0.856	0.033	−4.07	0.000	0.795–0.923
Children’s financial support	0.969	0.014	−2.09	0.037	0.942–0.998
Care expenditure	1.043	0.022	−1.99	0.046	1.001–1.087
Caregiver’s performance	0.249	0.049	−7.08	0.000	0.169–0.365
Share feelings with the caregiver	0.903	0.085	−1.08	0.280	0.750–1.087
Income sufficient for paying daily expenses	0.719	0.116	−2.05	0.041	0.524–0.986
Have real estate in own name	1.038	0.147	0.26	0.793	0.786–1.371
Public old age pension	1.086	0.133	0.68	0.499	0.855–1.380

**Note:** SE represent standard error.

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
