# Peer review of "Influencing Factors of Undermet Care Needs of the Chinese Disabled Oldest Old People When Their Children Are Both Caregivers and Older People: A Cross-Sectional Study"

_healthcare, 2020, doi:10.3390/healthcare8040365_

Round 1

Reviewer 1 Report

Authors might consider the suggestions below to further improve the manuscript.

(1) Authors are suggested to treat the variable–“Education”–as a categorical variable;

(2) Authors are advised to re-assign the variable–“Caregiver’s performance”, merging the categories of “Unwilling” and “Without patience” into one, given that there are only 8 cases under the category of “Unwilling”, which are generally inadequate for regression analysis and might lead to biased results.

(3) It would be better to adjust the variable–“Telling thoughts with caregiver” as well, merging the two categories of “Sometimes” and “Often” into one.

(4) It is also advised to break “frailty index” into eight parts. This would help authors better illustrate the effect of each part on the dependent variable.

(5) When operating binary logistic regression, authors are suggested to transform all categorical variables into dumb variables. This would facilitate better explanations of the regression results.

(6) Sociodemographic characteristics of caregivers need to be taken into account in the regressions.

Author Response

Dear Editors and Reviewers,

Thank you for your letter and for the reviewers’ comments concerning our manuscript entitled “Influencing Factors of Undermet Care Needs of the Chinese Disabled Oldest Old People When Their Children are Both Caregivers and Elderly People: A Cross-Sectional Study” (ID: healthcare-903314). Those comments are all valuable and very helpful for revising and improving our paper, and simultaneously they are of great significance to give instructions to our research. We have studied the comments carefully and made revisions which we hope to meet with approval. Revised portions are marked in red in the original paper. The main revisions on this paper, along with our responses to the reviewer’s comments, are as follows:

Point 1: Authors are suggested to treat the variable–“Education”–as a categorical variable

Response 1:

We are grateful to the reviewer for his/her suggestion. The variable – “Education” is treated as a categorial variable in the new manuscript which can be seen in Table 1 and Table 2.

Point 2: Authors are advised to re-assign the variable–“Caregiver’s performance”, merging the categories of “Unwilling” and “Without patience” into one, given that there are only 8 cases under the category of “Unwilling”, which are generally inadequate for regression analysis and might lead to biased results.

Response 2:

   We thank for the reviewer’s suggestion. The variable – “Caregiver’s performance” is re-assigned in the new manuscript which can be seen in Table 1.

Point 3: It would be better to adjust the variable – “Telling thoughts with caregiver” as well, merging the two categories of “Sometimes” and “Often” into one.

Response 3:

We agree with the reviewer’s suggestions and the variable – “Telling thoughts with caregiver” is adjusted, merging the two categories of “Sometimes” and “Often” into one in the new manuscript (see in Table 1).

Point 4: It is also advised to break “frailty index” into eight parts. This would help authors better illustrate the effect of each part on the dependent variable.

Response 4:

We agree with the reviewer’s suggestion. However, based on the research goals and framework of our study, we think that a detailed study of each parts of “frailty index” is not the focus of our article. If it is added, it will cause a sense of distraction.

First, the frailty index is a comprehensive index of elderly health, which can comprehensively reflect the health status of the Chinese disabled oldest old people. And it is more comprehensive and efficient than any other internal single parts in reflecting the health status of the oldest old people (Rockwood et al., 1994; Zeng et al., 2008; Gu et al., 2009).

Second, because our study examines the influencing factors of the undermet care needs in the context of the family care model in China. We want to focus more on exploring the influencing factor about cultural characteristics and traditional customs of care in Chinese oldest old people. Therefore, the influencing factors about family endowments, intergenerational relationships, and bargaining power of the oldest old people, not the health status is the focus of our study. Of course, we admit that the health status of the Chinese disabled oldest old people is very important, and we also use the frailty index to comprehensively reflect it.

Finally, our future research can further explore the impact of health factors on the undermet care needs by using each part of the frailty index, respectively. The suggestion has significant reference value for our future research. We are very grateful for the reviewers’ suggestions.

List of related references are as follows:

Rockwood, K.; Fox, R.A.; Stolee, P. Frailty in elderly people: an evolving concept. CMAJ 1994, 150, 489-495.

Gu, d.; Dupre, M.E.; Sautter, J. Frailty and mortality among Chinese at advanced ages. J Gerontol B Psychol Sci Soc Sci 2009, 64, 279-289.

Zeng, Y. Reliability of Age Reporting among the Chinese Oldest-old in the CLHLS Data Sets. Demographic Methods & Population Analysis. 2008, 20, 61-78.

Point 5: When operating binary logistic regression, authors are suggested to transform all categorical variables into dumb variables. This would facilitate better explanations of the regression results.

Response 5:

     We agree with the reviewer’s suggestion. However, there were two points that could not be achieved in our study. First, the categorical variables in regression analysis were in line with the classification method in the original questionnaire of CLHLS. Also, this classification method was widely used in binary logistic regression in the previous studies researched with CLHLS data (Gu et al., 2019; Zhang et al., 2019; Li et al., 2017). It was uncertain the dumb variables were applicable in the regression analysis, as there was little reference. Second, some categorical variables, such as “Share feelings with the caregiver” described the frequency and were not suitable to transform into dumb variables. Because the present classification was more refined, if forced to become a dummy variable, it would cause information loss. Last, but also the important, there were no big difference in the regression results by using these two different methods.

List of related references are as follows:

Gu, L.J.; Cheng, Y.; Phillips, D.R.; Rosenberg, M. Understanding the Wellbeing of the Oldest-Old in China: A Study of Socio-Economic and Geographical Variations Based on CLHLS Data. Int J Environ Res Public Health. 2019, 16, 601. doi: 10.3390/ijerph16040601.

Zhang, Q.L.; Wu, Y.L.; Han, T.K.; Liu, E.P. Changes in Cognitive Function and Risk Factors for Cognitive Impairment of the Elderly in China: 2005-2014. Int J Environ Res Public Health. 2019, 16, 2847. doi: 10.3390/ijerph16162847.

Li, J.; Wu, B.; Selbæk, G.; Krokstad, S.; Helvik, A.S. Factors associated with consumption of alcohol in older adults - a comparison between two cultures, China and Norway: the CLHLS and the HUNT-study. BMC Geriatr. 2017, 17, 172. doi: 10.1186/s12877-017-0562-9.

Point 6: Sociodemographic characteristics of caregivers need to be taken into account in the regressions.

Response 6:

We are very agreed with the reviewer’s suggestion. The sociodemographic characteristics of caregivers are very important in the process of analyzing undermet care needs of the Chinese oldest old people and need to be considered in the regressions. However due to the structure of the questionnaire in CLHLS, there is little information on the caregiver’s characteristics, and we also add this point as one limitation of our research. Of course, this suggestion also provides a good reference value for our future research. We thank the reviewers of his/her suggestions.

Reviewer 2 Report

The authors present and interesting paper regarding the factors that predict undermet needs in Chinese disabled oldest old people. Please consider these minor comments.

Please clarify the acronyms RMB in the introduction section.

Please include in the introduction section a reference to the COVID-19 situation and how it has impacted on the care of elderly people in China.

In table 1 why did you use p-values? If you are just describing the sample I don’t understand why you made comparisons. Please clarify. I also recommend changing the distribution of the table to made them more clear. 

Please include the psychometric properties of the instruments described in the measures section.

Why did you state that higher care expenditure (with and OR= 1.044) is inversely associated? Please clarify this in the result section.

Author Response

Dear Editors and Reviewers,

Thank you for your letter and for the reviewers’ comments concerning our manuscript entitled “Influencing Factors of Undermet Care Needs of the Chinese Disabled Oldest Old People When Their Children are Both Caregivers and Elderly People: A Cross-Sectional Study” (ID: healthcare-903314).Those comments are all valuable and very helpful for revising and improving our paper, and simultaneously they are of great significance to give instructions to our research. We have studied the comments carefully and made revisions which we hope to meet with approval. Revised portions are marked in red in the original paper. The main revisions on this paper, along with our responses to the reviewer’s comments, are as follows:

Point 1: Please clarify the acronyms RMB in the introduction section.

Response 1:

We apologized to the reviewer for missing the explanation of the acronyms RMB. RMB is the short of Ren Min Bi, also known as Chinese Yuan. We have added this explanation and converted the currency unit from RMB to U.S. dollars to make it easier to understand in the new manuscript (See line 65-68).

Point 2: Please include in the introduction section a reference to the COVID-19 situation and how it has impacted on the care of elderly people in China.

Response 2:

We agree with the reviewers’ suggestions and add some information about the impact of COVID-19 on the care of Chinese older people. The details are as follows:

The outbreak of the COVID-19 had a significant impact on the care of Chinese older people, especially the family care model. The Ministry of Civil Affairs, in charge of older people’s care services, had issued some policy support and guidance for institutional older people care, such as “Guidelines for Prevention and Control of COVID-19 Epidemic High-risk Areas and Infected Older People Care Institutions”. However, no corresponding policy support measures had been introduced for older people in family care. The impacts on older people cared by the family members included, but are not limited to. The older adults suffering from chronic diseases could not get medicines in time. Also, the older people were in a state of panic, restricted at home due to their inability to use information equipment proficiently.

Point 3: In table 1 why did you use p-values? If you are just describing the sample I don’t understand why you made comparisons. Please clarify. I also recommend changing the distribution of the table to made them more clear.

Response 3:

We appreciate the comments of the reviewers. We aimed to describe the characteristics of the sample and did not involve the content of comparison in Table 1. Therefore, the P-value was redundant here and should be removed. We realized that we made a mistake in the original manuscript, and removed the P-value in the new manuscript (See in Table 1). We are very grateful to the reviewers for their suggestions.

The distributation of Table 1 has referred to the latest articles (Ansa et al., 2020; Fu et al., 2020; Park et al., 2020) published by Healthcare.

List of related references are as follows:

  1. Ansa, B.E.; Zechariah, S.; Gates, A.M.; Johnson, S.W.; Heboyan, V.; Leo, G.D. Attitudes and Behavior towards Interprofessional Collaboration among Healthcare Professionals in a Large Academic Medical Center. Healthcare. 2020, 8, 316.
  2. Fu, L.P.; Wang, Y.H.; He, L.P. Factors Associated with Healthy Ageing, Healthy Status and Community Nursing Needs among the Rural Elderly in an Empty Nest Family: Results from the China Health and Retirement Longitudinal Study. Healthcare. 2020, 8, 317.
  3. Park, S.J.; Kim, S.H.; Kim, S.H. Effects of Thoracic Mobilization and Extension Exercise on Thoracic Alignment and Shoulder Function in Patients with Subacromial Impingement Syndrome: A Randomized Controlled Pilot Study. Healthcare. 2020, 8, 316.

Point 4: Please include the psychometric properties of the instruments described in the measures section.

Response 4:

We agree with the reviewer’s suggestion. The “psychometric properties of the instruments” mentioned by the reviewer is mainly reflected in the cognitive function part of the frailty index. In the CLHLS database, the modified MMSE (Mini-mental State Examination), abbreviated as CMMSE (Chinese Mini-mental State Examination), is mainly used to measure the cognitive function of the older people in China. Regarding the suitability of CMMSE in measuring the cognitive function of the older people in China, the official website of CLHLS has made a detailed explanation( For detailed information, please refer to the website of the CLHLS project of Peking University and Duke University: https://opendata.pku.edu.cn/dataset.xhtmlpersistentId=doi:10.18170/DVN/XRV2WN or https://sites.duke.edu/centerforaging/programs/chinese-longitudinal-healthy-longevity-survey-clhls/), and have been applied in some previous studies (An & Liu, 2016; Zeng et al., 2017 Gao et al., 2018; Zhang et al., 2019).

Related references:

  1. An, R., & Liu, G. G.. Cognitive impairment and mortality among the oldest-old Chinese. International Journal of Geriatric Psychiatry. 2016, 31, 1345-1353.
  2. Zeng, Y., Feng, Q. S., Hesketh, T., Christensen, K., & Vaupel, J. W. Survival, disabilities in activities of daily living, and physical and cognitive functioning among the oldest-old in China: A cohort study. The Lancet. 2017, 389, 1619-1629.
  3. Gao, M., Sa, Z., Li, Y., Zhang, W., Tian, D., Zhang, S., & Gu, L. Does social participation reduce the risk of functional disability among older adults in China? A survival analysis using the 2005-2011 waves of the CLHLS data. BMC Geriatrics. 2018, 18(1), 224.
  4. Zhang Q, Wu Y, Han T, Liu E. Changes in Cognitive Function and Risk Factors for Cognitive Impairment of the Elderly in China: 2005-2014. International Journal of Environmental Research and Public Health. 2019, 16(16), 2847.

Point 5: Why did you state that higher care expenditure (with and OR = 1.044) is inversely associated? Please clarify this in the result section.

Response 5:

    We are grateful to the reviewer for his/her suggestion. To a certain extent, the care expenditure represents the explicit cost in the process of family care for Chinese disabled oldest old people. The more care expenditure, the more resources put into care process, and the lower the undermet care needs of the Chinese disabled oldest old people. We also added the explanation in the result section. (See line 265-268)

Reviewer 3 Report

This paper reports the findings from a cross-sectional study , a secondary analysis, on unfulfilled care needs of oldest old people living with disability and chronic conditions in China and for whom their primary carer is an adult child aged 60 years and over. As such, this is in scope for ‘Healthcare’ readership and should be of interest.

Background

Global ageing demography is considered and from within China particularly, in relation to the increase and projected increase, in the older old population, aged 80 years and older. The authors also offer current evidence to illustrate how in general, this population in China is more likely to be living with poor health and disability and how, particularly from within older generations, there is still a cultural expectation of filial piety. With the oldest old, it is likely that their adult children and children-in-law, those most commonly taking up this familial care role, are likely to be ageing themselves, 60 years and over. The limitations of  global studies that have in general, explored   the influencing factors for under met care needs of oldest old people,  are presented and justify the need for this study, in particular, the lack of a Chinese context in relation to its rapidly ageing demography, limited samples and lack of a focus on family carer aged 60 years and over.

Methodology

The study’s secondary analysis, data source, the 2018 cross sectional data from the Chinese Longitudinal Healthy Longevity Survey (CLHLS), is well explained, and its population, well described.  From within the CLHS survey context , measures focusing on what is meant by unfulfilled care needs: ‘under met care needs’ and ‘influencing factors of under met care needs’ are well defined, as are ‘Disabled Oldest Old People’ and ‘Elderly [sic] children’ . Descriptive statistical analysis is congruent with the study’s cross-sectional design and tabular presentation is welcome.

I may have missed this but there doesn’t seem to be a reference to ethical approvals – I understand that this study only carried out secondary data analysis of existing data, but I would have expected either host institutional ethical approval being sought and granted, or an author explanation as to why this was not needed. Apologies if ethical approval is indeed recorded, if not in this manuscript, perhaps within other submission documentation.

Results

Living in rural areas, having a poor health status as indicated by a higher frailty index and a lower annual household income, were all associated with having more under met care needs, these corroborated by previous studies’ findings. Gender, age, education level and marital status were not significantly associated with under met care needs of the disabled oldest old people, despite the mixed results of previous studies.  More children’s financial support, care expenditure and better caregiver’s performance were inversely associated with having more under met care needs, again supported by previous studies’ findings. What constitutes ‘better caregiver’s performance’ is unclear. Economic bargaining power, having enough money to pay for daily expenses,(was significantly inversely associated with higher  odds for under met care needs of the disabled oldest old people, although both  having real estate in one’s own name and having public old age insurance, were not significantly associated.

 Frequency of ‘‘better caregiver’s performance’, a measure used within the CLHLS survey and described as one of the ‘influencing factors of under met care needs’ was not associated with under-met care needs. The authors offered the following explanation: “[It] may be that regular emotional communication with elderly children does not necessarily/always bring about improvements in care among disabled oldest old people.” [p.7] I would suggest that for an international readership it would be helpful to offer an example of what is meant by ‘sharing thoughts with the caregiver’? I’m assuming that this is very much in the affective domain, perhaps alluding to the ‘cared for’ being able to freely express their opinion on how their needs are being met? I realise that such measures as ‘better caregiver’s performance’ and frequency of ‘sharing thoughts with the caregiver ‘, are not nuanced from within a survey, but perhaps this should be drawn out a little more, either in the discussion or the limitations sections? 

Discussion/Conclusion

Results are discussed with some possible further explanation offered.  Whilst there is acknowledgement that filial piety is still prevalent, there is acknowledgement of wider national cultural and economic change such as an increasing geographical mobility of the population, the rising female employment rate and the weakening of traditional values. Such change suggests that available family caregivers are likely to decline and this is exacerbated by an ageing demography. Policy implications of these findings are well articulated as are study limitations

Overall, this is a very well presented and articulated paper. I have very minor requests, clarification of ethical status and some further explanation/discussion focusing on what is meant by ‘better caregiver’s performance’  and  ‘sharing thoughts with the caregiver ‘,    – these latter just to give a little more contest to the findings – would be most helpful.

Author Response

Dear Editors and Reviewers,

Thank you for your letter and for the reviewers’ comments concerning our manuscript entitled “Influencing Factors of Undermet Care Needs of the Chinese Disabled Oldest Old People When Their Children are Both Caregivers and Elderly People: A Cross-Sectional Study” (ID: healthcare-903314).Those comments are all valuable and very helpful for revising and improving our paper, and simultaneously they are of great significance to give instructions to our research. We have studied the comments carefully and made revisions which we hope to meet with approval. Revised portions are marked in red in the original paper. The main revisions on this paper, along with our responses to the reviewer’s comments, are as follows:

Point 1 and 2: Overall, this is a very well presented and articulated paper. I have very minor requests, clarification of ethical status and some further explanation/discussion focusing on what is meant by ‘better caregiver’s performance’ and ‘sharing thoughts with the caregiver’, – these latter just to give a little more contest to the findings – would be most helpful.

Response 1:

We agree with the reviewer’s suggestion and add information regarding ethical approval of this study.

Ethical approval: The study was approved by the institutional review board at Centre for Social Security Studies of Wuhan University.

Response 2:

    We are very grateful to the reviewer for his/her appreciation of the manuscript and suggestions.

Among them, regarding the “caregiver’s performance”, we think that it is a very important factor that affects the undermet care needs of the Chinese oldest old people. Of course, this is a more comprehensive concept, which includes both the objective care behaviours of the caregiver and the subjective attitude of the caregiver in the care provision process. Better care performance means that the caregiver’s behaviours are more in line with the care needs of the older people, and there are more love and positive emotions involved in the care process.

The corresponding revision in the manuscript is: As for the care performance, it includes both the caregiver’s objective care behaviour and the caregiver’s subjective attitude. So, it is a more comprehensive concept. Better care performance means that the caregiver’s behaviours are more in line with the care needs of the older people, and there are more love and positive emotions involved in the care process.

When it comes to “telling thoughts with the caregiver”, we must admit to the reviewer that there is a mistake in our expression. Its original meaning should be “sharing feelings with the caregiver”. It goes without saying that this factor is a measure of intergenerational relationships and their impact on the undermet care needs of the older people from the perspective of emotional communication or interaction. Therefore, we replaced the expression “telling thoughts with the caregiver” with “sharing feelings with the caregiver” to better convey the emotional interaction between the oldest old people and their children.

Reviewer 4 Report

This paper examined factors associated with unmet need among a large cohort of older adults in China. The paper focused on older adults who were in receipt of care from their children who were themselves aged 60 years and older.

It is well written and addresses an important and growing problem for many ageing societies.

An important point to note is that the word 'elderly' is considered derogatory so please replace this throughout the manuscript. Acceptable terms include 'older adult(s)', 'older person(s)' etc. There are lots of editorials available about this issue.

Introduction:
The authors do a good job of describing the care model in China. In particular, the second paragraph on page 2 (L58 to 78) does an excellent job of succinctly describing a complex phenomenon.

The research question is clear and well justified.

Methods:

Page 3 L 122 - Can the average years of education really be 1.4 years? It would be helpful to see the range here more so than the standard deviation.

Table 1. What exactly are the p values showing here? They don't make any sense.

Please define 'family endowment' in this context.

Page 5 L 186. As this is a two tailed test, doesn't 0.05 on both sides imply that the alpha level would then be 0.90. Regardless, it isn't necessary to state this either way.

Where the data weighted? If not, how representative is the sample of the wider population. There is mention of this in places but it could be expanded. 

Results:

Might there be an issue with confounding for the independent variable 'care giving performance'. By this I mean, won't care giving performance be necessarily reported as poor where there is unmet need and vice versa. What happens to the the other estimates if this item is excluded from the logistic regression model?

Discussion:
Andersen’s behavioral model of health services use is introduced for the first time in the discussion. This should first be mentioned, and described, in the introduction. Also the citation for this theory, which I think is Fortin et al. should come after the first mention.

The discussion of the non-significant association of the number of cohabitants might also consider the some of the reasons as to why multiple generations live in the same household. For example, it might be that the caregivers in this situation have increased needs themselves or lack their own resources (human or material capital) to live independently. Also, it may be that there are increased expectations assigned to caergivers who live in the family home.

It is good that the authors mention as a limitation the fact that the characteristics of the caregivers are not considered. One would guess that things like the caregivers own health status and other caring responsibilities play a major role.

Minor points:
Page 2 L59 - Not clear what 'inertial old-age care model' means.
Page 2 L73 - The quotation marks at "living in the nursing home" are unnecessary. Just replace the with a.
The layout of Table 1 could also improved as the centering of cells makes it difficult to read.

Table 2. There are no means or SD reported so the authors can remove this from the table heading. I would also like to see 95% confidence intervals reported to get a better sense of the precision of the estimates. For example, I would guess that the confidence intervals for the 'have spouse' group will be quite wide.

Author Response

Dear Editors and Reviewers,

Thank you for your letter and for the reviewers’ comments concerning our manuscript entitled “Influencing Factors of Undermet Care Needs of the Chinese Disabled Oldest Old People When Their Children are Both Caregivers and Elderly People: A Cross-Sectional Study” (ID: healthcare-903314).Those comments are all valuable and very helpful for revising and improving our paper, and simultaneously they are of great significance to give instructions to our research. We have studied the comments carefully and made revisions which we hope to meet with approval. Revised portions are marked in red in the original paper. The main revisions on this paper, along with our responses to the reviewer’s comments, are as follows:

Point 1: An important point to note is that the word 'elderly' is considered derogatory so please replace this throughout the manuscript. Acceptable terms include 'older adult(s)', 'older person(s)' etc. There are lots of editorials available about this issue.

Response 1:

We are very grateful to the reviewers for his/her suggestion. The whole manuscript has been modified after adopting the reviewer’s suggestion.

Point 2: Page 3 L 122 - Can the average years of education really be 1.4 years? It would be helpful to see the range here more so than the standard deviation.

Response 2:

The current Chinese oldest old people were born before founding of the People’s Republic of China, when there were at wars and fewer opportunities for formal education. At that time, many people didn’t go to school and only few rich people go to the private school. That’s the reason why the average education years in current Chinese oldest old people was only 1.4.

The range of education years can indeed provide some information on education level of Chinese oldest old people, however in order to be consistent with other variables, we chose to report the mean and standard deviation of education years. And we believe that the mean and standard deviation can better reflect the overall characteristics of the variable than the single range.

Point 3: Table 1. What exactly are the p-values showing here? They don't make any sense.

Response 3:

     We appreciate the comments of the reviewers. We aimed to describe the characteristics of the sample and do not involve the content of comparison in Table 1. Therefore, the P-value is redundant here and should be removed. We realized that we made a mistake in the original manuscript, and removed the P-value in the new manuscript (See in Table 1). We are very grateful to the reviewers for their suggestions.

Point 4: Please define 'family endowment' in this context.

Response 4:

We agree with the reviewer’s suggestion. The corresponding supplementary content is as follows:

Family endowment also called family resource, include the human resources, economic resources, material resources, and social network resources which is the family actually or potentially own. In our research, the measurement of family endowment mainly included the number of cohabitating family members and annual household income.

Point 5: Page 5 L 186. As this is a two tailed test, doesn't 0.05 on both sides imply that the alpha level would then be 0.90. Regardless, it isn't necessary to state this either way.

Response 5:

We apologized to the reviewers for making a mistake in introducing the method of regression. The test level was 0.025 not 0.05 on both sides. We have removed this content in the new manuscript after adopting the reviewer’s suggestion.

Point 6: Where the data weighted? If not, how representative is the sample of the wider population. There is mention of this in places but it could be expanded.

Response 6:

    We appreciate the comments of the reviewers. We apologize for ignoring the description of the sampling weight in original manuscript. Based on the recommendations of the reviewers, we have added the content of sampling weight and relevant reference in the Date Source and Study Population section. (See line 112-114)

Given that there are fewer persons at more advanced ages, and fewer males than females, in order to avoid random fluctuation errors caused by the small sample size of the elderly, especially the males, the CLHLS does not take proportional sampling, and over-samples extremely elderly and male elderly instead. Detailed information about the sampling weight in CLHLS can be found in previous studies (Zeng et al., 2001; Zeng et al., 2008) and following websites:

https://opendata.pku.edu.cn/dataset.xhtml?persistentId=doi:10.18170/DVN/XRV2WN

List of related references are as follows:

  1. Zeng, Y., Vaupel, J.W., Xiao, Z., Zhang, C., Liu, Y., 2001. The healthy longevity survey and the active life expectancy of the oldest old in China. Population: An English Selection. 13, 95-116.
  2. Zeng, Y., Poston, D.L., Vlosky, D.A., Gu, D.N., 2008. Healthy longevity in China: Demographic, socioeconomic, and psychological dimensions. Dordrecht, The Netherlands: Springer Publisher.

Point 7: Might there be an issue with confounding for the independent variable 'care giving performance'. By this I mean, won't care giving performance be necessarily reported as poor where there is unmet need and vice versa. What happens to the other estimates if this item is excluded from the logistic regression model?

Response 7:

We distinguished between the two situations of “Unmet care needs” and “Undermet care needs” in our study. Followed by previous studies (Kennedy, 2001; Peng et al., 2015), absent and insufficient assistance in personal activities of ADL were named unmet and undermet care needs, respectively. In our study, the care of Chinese oldest old people was provided by their children or children in-law, which showed the care was available. Therefore, we used the term “undermet care needs” to define insufficient assistance in personal activities of ADL among the disabled oldest old people. We wanted to clarify that our research object was “undermet care needs” not “unmet care needs”. Although there is undermet care need, the care of Chinese disabled oldest old people was available and provided by their children. It is absolute necessary to study care giving performance, as it is a comprehensive evaluation of care and should be included in the analysis of undermet care needs. Besides, the information of caregiver in our study is very few and the “caregiver’s performance” is a very important variable which cannot be ignored.

We also excluded the variable “caregiver’s performance” from the logistic regression model, and other variables changed very little which could be ignored. The regression results are as follows:

Before:
Logistic regression

Variables

Odds Ratio

SE

z-Values

p-values

95% CI

Sig

 A1(Sex)

1.195

0.166

1.28

0.201

0.910-1.568

 A2(Age)

0.988

0.011

-1.12

0.261

0.967-1.009

 A3(Education)

0.911

0.096

-0.89

0.376

0.742-1.119

 A4(Marital status)

0.767

0.203

-1.00

0.316

0.457-1.288

 A5(Place of residence)

1.308

0.096

3.65

0.000

1.132-1.510

***

 A6(Frailty index)

1.102

0.014

7.43

0.000

1.074-1.130

***

 B1(Number of cohabitating family members)

1.011

0.033

0.32

0.751

0.947-1.078

 B2(Annual household income)

0.856

0.033

-4.07

0.000

0.795-0.923

***

 C1(Children’s financial support)

0.969

0.014

-2.09

0.037

0.942-0.998

**

 C2(Care expenditure)

1.043

0.022

1.99

0.046

1.001-1.087

**

 C3(Caregiver’s performance)

0.249

0.049

-7.08

0.000

0.169-0.365

***

 C4(Share feelings with the caregiver)

0.903

0.085

-1.08

0.280

0.750-1.087

 D1(Income sufficient for paying daily expenses)

0.719

0.116

-2.05

0.041

0.524-0.986

**

 D2(Have real estate in own name)

1.038

0.147

0.26

0.793

0.786-1.371

 D3(Public old age pension)

1.086

0.133

0.68

0.499

0.855-1.380

 Constant

10.966

13.650

1.92

0.054

0.956-125.769

*

Mean dependent var

0.495

SD dependent var

Pseudo r-squared

0.100

Number of obs 

Chi-square 

222.446

Prob > chi2

Akaike crit. (AIC)

2035.783

Bayesian crit. (BIC)

*** p<0.01, ** p<0.05, * p<0.1

After excluding the variable “caregiver’s performance”

Logistic regression

Variables

Odds Ratio

S.E.

z-value

p-value

95% CI

Sig

 A1(Sex)

1.155

0.158

1.06

0.289

0.884-1.510

 A2(Age)

0.992

0.010

-0.78

0.438

0.972-1.013

 A3(Education)

0.921

0.095

-0.80

0.425

0.753-1.127

 A4(Marital status)

0.806

0.209

-0.83

0.405

0.485-1.340

 A5(Place of residence)

1.336

0.096

4.02

0.000

1.160-1.538

***

 A6(Frailty index)

1.108

0.014

7.95

0.000

1.080-1.136

***

 B1(Number of cohabitating family members)

1.006

0.033

0.18

0.860

0.944-1.072

 B2(Annual household income)

0.861

0.033

-3.98

0.000

0.799-0.927

***

 C1(Children’s financial support)

0.969

0.014

-2.13

0.033

0.942-0.998

**

 C2(Care expenditure)

1.049

0.022

2.31

0.021

1.007-1.092

**

 C4(Share feelings with the caregiver)

0.881

0.082

-1.36

0.173

0.734-1.057

 D1(Income sufficient for paying daily expenses)

0.637

0.100

-2.89

0.004

0.469-0.865

***

 D2(Have real estate in own name)

1.035

0.144

0.25

0.806

0.787-1.360

 D3(Public old age pension)

1.074

0.129

0.59

0.553

0.849-1.359

Constant

2.071

2.486

0.61

0.544

0.197-21.767

Mean dependent var

0.495

SD dependent var

Pseudo r-squared

0.073

Number of obs 

Chi-square 

163.550

Prob > chi2

Akaike crit. (AIC)

2092.679

Bayesian crit. (BIC)

*** p<0.01, ** p<0.05, * p<0.1

List of related references are as follows:

  1. Kennedy, J. Unmet and undermet need for activities of daily living and instrumental activities of daily living assistance among adults with disabilities: Estimates from the 1994 and 1995 disability follow-back surveys. Med. Care. 2001 39, 1305–1312.
  2. Peng, R.; Wu,B.; Ling, L. Undermet needs for assistance in personal activities of daily living among community-dwelling oldest old in China from 2005 to 2008. Res. Aging 2015, 37, 148-170.

Point 8: Andersen’s behavioral model of health services use is introduced for the first time in the discussion. This should first be mentioned, and described, in the introduction. Also the citation for this theory, which I think is Fortin et al. should come after the first mention.

Response 8:

We agree with the reviewer’s suggestion. Andersen’s behavioral model of health services use should be introduced in the first part to further highlight the innovations of this study compared to previous studies. We also believe that Fortin et al (2018) has a more direct and important reference value for our research and should be mentioned in the first part. Based on the reviewer’s suggestions, we made corresponding changes. The details are as follows:

Third, previous studies on undermet care needs of older people mostly used Andersen’s behavioral model of health services use (Fortin et al., 2018; Desai et al., 2001; Zhu, 2015), which may not fully consider the cultural characteristics and traditional customs of Chinese older people care.

Point 9: The discussion of the non-significant association of the number of cohabitants might also consider the some of the reasons as to why multiple generations live in the same household. For example, it might be that the caregivers in this situation have increased needs themselves or lack their own resources (human or material capital) to live independently. Also, it may be that there are increased expectations assigned to caregivers who live in the family home.

Response 9:

We agree with the theoretical mechanism provided by the reviewer about non-significant association of the number of cohabitants with undermet care needs of older people. Therefore, we have added this content in the revision manuscript:

Another explanation is because multiple generations live together, the care needs of the older people may not be able to be satisfied when family resources are limited and the older people have less power in the distribution of family resources.

Point 10: It is good that the authors mention as a limitation the fact that the characteristics of the caregivers are not considered. One would guess that things like the caregiver’s own health status and other caring responsibilities play a major role.

Response 10:

We strongly agree with the reviewer’s suggestion. The characteristics of the caregivers, such as education, health status, and other caring responsibilities are very important in studying the undermet care needs of the Chinese disabled oldest old people, and should be considered. However, due to the constraints of the data, we lack data on this information and we also added it in the limitation. Our future study will further explore the characteristics of the caregivers in the undermet care needs. Of course, this suggestion provides a good reference value for our future research. We thank the reviewers of his/her suggestions.

Point 11: Page 2 L59 - Not clear what 'inertial old-age care model' means.

Response 11:

The “inertial old-age care model” mentioned in our manuscript refers to the strong inertia of China’s family care model. For thousands of years, taking care of parents by children has been the main mode of care for the elderly in China. Although China is currently undergoing rapid economic and social changes, cultural transformation, from a traditional society to a modern society. Correspondingly, there has also been a model of institutional care for the elderly and the government providing elderly care services. However, the mode of taking care of parents by children is still the main one, with more than 90% of elderly care providers being their children.

Point 12: Page 2 L73 - The quotation marks at "living in the nursing home" are unnecessary. Just replace the with a.

Response 12:

    It has been modified in the new manuscript which can be seen in Line 73.

Point 13: The layout of Table 1 could also improved as the centering of cells makes it difficult to read.

Response 13:

The layout of Table 1 has referred to the latest articles (Ansa et al., 2020; Fu et al., 2020; Park et al., 2020) published by Healthcare. In these articles, the cells were all centered.

List of related references are as follows:

  1. Ansa, B.E.; Zechariah, S.; Gates, A.M.; Johnson, S.W.; Heboyan, V.; Leo, G.D. Attitudes and Behavior towards Interprofessional Collaboration among Healthcare Professionals in a Large Academic Medical Center. Healthcare. 2020, 8, 316.
  2. Fu, L.P.; Wang, Y.H.; He, L.P. Factors Associated with Healthy Ageing, Healthy Status and Community Nursing Needs among the Rural Elderly in an Empty Nest Family: Results from the China Health and Retirement Longitudinal Study. Healthcare. 2020, 8, 317.
  3. Park, S.J.; Kim, S.H.; Kim, S.H. Effects of Thoracic Mobilization and Extension Exercise on Thoracic Alignment and Shoulder Function in Patients with Subacromial Impingement Syndrome: A Randomized Controlled Pilot Study. Healthcare. 2020, 8, 316.

Point 14: Table 2. There are no means or SD reported so the authors can remove this from the table heading. I would also like to see 95% confidence intervals reported to get a better sense of the precision of the estimates. For example, I would guess that the confidence intervals for the 'have spouse' group will be quite wide.

Response 14:

We did report means and SD in Table 2, thereby this content could not be removed from the table heading. Besides, the content and layout of Table 2 has referred to the articles (Kim, 2017; Abreu-Sánchez, 2020; Kim, 2020) published by Healthcare. These articles did not report the 95% confidence intervals in the parts of analyzing demographic characteristics of the research subjects. If the reviewer insists that reporting the 95% confidence intervals is very necessary in the Table 2, we will later add this content.

List of related references are as follows:

  1. Kim, K. Use of Clinical Preventive Service and Related Factors in Middle-Aged Postmenopausal Women in Korea. Healthcare. 2020, 8, 83.
  2. Kim, C.B.; Yoon, S.J.; Ko, J. Economic Activity and Health Conditions in Adults Aged 65 Years and Older: Findings of the Korean National Longitudinal Study on Aging. Healthcare.2017, 5(4), 63.
  3. Abreu-Sánchez, A.; Parra-Fernández, M.L.; Onieva-Zafra, M.D.; Ramos-Pichardo, J.D.; Fernández-Martínez, E. Type of Dysmenorrhea, Menstrual Characteristics and Symptoms in Nursing Students in Southern Spain. Healthcare. 2020, 8(3), 302.
